# Transcriptome Analysis Reveals Gemykibivirus Infection Induces Mitochondrial DNA Release in HEK293T Cells

**DOI:** 10.3390/v17101331

**Published:** 2025-09-30

**Authors:** Runbo Yang, Hao Yan, Yifan Wang, Wenqing Yang, Jianru Qin

**Affiliations:** 1College of Life Science, Henan Normal University, Xinxiang 453007, China; yymala@163.com (R.Y.); 18348041068@163.com (H.Y.); 18737364146@163.com (W.Y.); 2State Key Laboratory of Antiviral Drugs, Henan Normal University, Xinxiang 453007, China; 8040@163.com; 3Pingyuan Laboratory, Xinxiang 453007, China

**Keywords:** gemykibivirus, RNA sequencing, mitochondrial dysfunction, Genomoviridae, emerging viruses

## Abstract

Gemykibivirus, an emerging single-stranded DNA (ssDNA) virus of the recently established genus in the family of Genomoviridae, had been discovered in human blood and cerebrospinal fluid and a variety of other body fluids. However, the molecular mechanisms of gemykibivirus entrance into the host cells and its pathogenicity remain poorly understood. To investigate the host response of gemykibivirus, we used an infectious clone of gemykibivirus previously established through molecular biology techniques to rescue virus in HEK293T cells and analyzed the changes in the host transcriptome during the infection period by RNA-Seq. Our findings indicate that gemykibivirus can both express viral proteins and accomplish replication, and high-throughput transcriptome analysis identified a total 1732 significantly different genes. Functional enrichment analysis of Gene Ontology (GO) and Kyoto Encyclopedia of Genes and Genomes (KEGG) pathways for differentially expressed genes (DEGs) showed gemykibivirus involving several important pathways, including MAPK signaling pathway, Chemical carcinogenesis-reactive oxygen species and Oxidative phosphorylation. Interestingly, mitochondrial DNA-encoded mRNAs exhibited varying levels of upregulation, suggesting that gemykibivirus may be involved in mitochondrial fission and the regulation of mitochondrial function. Subsequently, a series of experiments proved that gemykibivirus can lead an increase in mitochondrial DNA copy number, promote the release of mtDNA into the cytoplasm, enhance reactive oxygen species production and trigger other cellular antiviral responses. Overall, we lay a foundation for revealing the relationship between Gemykibivirus and human diseases through mitochondrial functional alterations.

## 1. Introduction

Gemykibivirus is a small circular DNA virus with an approximately 2.2 kB genome that is classified within the Genomoviridae family. It encodes a capsid protein (Cap) and replication-associated protein (Rep) [1]. Gemykibivirus are widely distributed in the environment and have been detected in untreated sewage and healthcare-associated aerosols in Chinese hospitals, plants, animals, insects, and humans [2,3,4,5,6]. Gemykibivirus has been recently detected in the oral and rectal swabs of bats, underscoring the potential risk of zoonotic transmission [7]. Among the Genomoviridae, Gemykibivirus is the most frequently detected in human samples, as identified by metagenomic sequencing. Furthermore, the virus has been detected in the blood samples of parenterally infected individuals and blood donors in Brazil [8], cerebrospinal fluid of patients with encephalitis, and blood of patients with HBV and HIV [9,10]. Notably, its detection in the lower respiratory tract of a patient with severe respiratory distress provided the first evidence suggesting a potential link between gemykibivirus and human disease [11].

Mitochondria, through oxidative phosphorylation, are the primary ATP source and are crucial in the antiviral response as well as in regulating programmed cell death [12]. Mitochondrial homeostasis is often disrupted during viral infection, resulting in a decreased mitochondrial membrane potential, production of reactive oxygen species (ROS), and the release of mitochondrial DNA into the cytoplasm [13].

Mitochondrial membrane damage, oxidative stress, or apoptosis can trigger mtDNA release and is recognized as a damage-associated molecular pattern (DAMP), which induces innate immune responses [14]. For example, it can be detected using cGAS, a cytosolic DNA sensor that catalyzes the cGAMP production and subsequently activates the STING pathway. This activation induces the production of type I interferons and pro-inflammatory cytokines, contributing to cellular immune responses and inflammation regulation [15].

Herein, we obtained the complete gemykibivirus genome from a patient sample and investigated the potential impact of gemykibivirus on human cells by transfecting a gemykibivirus-infectious clone into HEK293T cells for the first time to simulate gemykibivirus infection. Furthermore, we performed RNA sequencing (RNA-seq) analysis, which may provide new insights into its replication, pathogenic potential, and host–virus interactions. We investigated whether gemykibivirus mediates mitochondrial dysfunction and induces apoptosis to further explore the role of mitochondrial in immune response during infection.

## 2. Materials and Methods

### 2.1. Cells and Virus

Human embryonic kidney HEK293T (HEK293T) cells were obtained from the American Type Culture Collection (ATCC, Rockville, MD, USA) and cultured in Dulbecco’s Modified Eagle’s Medium (Gibco BRL, Grand Island, NY, USA) supplemented with 10% fetal bovine serum (HyClone, Logan, UT, USA) and 1% penicillin-streptomycin.

The gemykibivirus 2 CN-GZ1 strain was isolated from the sputum of an older woman with unexplained acute respiratory distress syndrome in Guangzhou, China, in February 2017. To construct the infectious clone, the full-length gemykibivirus genome was inserted into the pBluescript vector with EcoRI restriction sites at both termini. The recombinant plasmid was propagated in *E. coli* DH5α, and the viral genome was subsequently excised from the plasmid by EcoRI digestion and circularized with T4 DNA ligase (TaKaRa, Beijing, China). The ligation products were then purified using a DNA clean up kit (Omega Bio-Tek, Norcross, GA, USA), yielding a circular double-stranded DNA form of gemykibivirus that closely mimics its natural conformation. The infectious clone was transfected into HEK293T cells using Lipofectamine 3000 (Thermo Fisher Scientific, Waltham, MA, USA) to rescue the gemykibivirus.

### 2.2. Isolation of Total RNA and RNA-Seq

HEK293T cells were plated in 6-well cell culture plates. The cells were transfected with 300 ng per well of gemykibivirus infectious clone 24 h later, and the negative control group was transfected with the same volume of buffer used to dissolve the viral DNA. Two days later, the total RNA was extracted from the HEK293T cells using TRIzol (Invitrogen, Carlsbad, CA, USA) and treated with RNase-free DNase I to remove genomic DNA contamination. The high-quality RNA samples were subsequently submitted to the Sangon Biotech Co., Ltd. (Shanghai, China) for library preparation. Total RNA (1 μg per sample) was used to construct sequencing libraries with the VAHTS™ mRNA-seq V2 Library Prep Kit (Illumina^®^) (Thermo Fisher Scientific, Waltham, MA, USA) following the manufacturer’s protocol.

### 2.3. Processing of the Raw Sequence Data

The sequencing data were quality controlled and filtered using fastp (version 0.23.2), reads were removed if they were shorter than 20 nt, contained >15 Ns, had a base quality score < 5, or those in which >50% of the bases failed the quality filter. Clean reads were mapped to the Homo sapiens reference genome (GRCh38) retrieved from the National Center for Biotechnology Information (NCBI) database and mapped using HISAT2 (version 2.2.1) with default parameters. The gene expression values of the transcripts were computed using featureCounts (version 2.0.1). DESeq2 (version 1.48.1) was used to determine differentially expressed genes (DEGs) between two samples. Genes were considered significantly different if *q*-value ≤ 0.05 and |FoldChange| ≥ 1.

### 2.4. Functional Analysis of DEGs

Functional enrichment analyses, including Gene Ontology (GO) and Kyoto Encyclopedia of Genes and Genomes (KEGG) analyses, were preformed to identify which DEGs were markedly enriched in the GO terms or metabolic pathways. GO is an international standard classification system for gene function. DEGs were mapped to the GO terms (biological functions) in the database, the number of genes in each term is calculated, and a hypergeometric test was conducted to identify the substantially enriched GO terms in the gene list from the background of the reference gene list. The KEGG database is a public pathways database. The KEGG pathway analysis identifies markedly enriched metabolic pathways or signal transduction pathways enriched in the DEGs compared to a reference gene background using the hypergeometric test. The GO terms and KEGG pathways with a false discovery rate (*q*-value) < 0.05 were considered significantly altered.

### 2.5. Protein Extraction and Western Blot Analysis

Total protein was extracted using radioimmunoprecipitation assay lysis buffer (Beyotime, Shanghai, China), following the manufacturer’s protocol. The extracted proteins were separated using 10% sodium dodecyl sulfate-polyacrylamide gel electrophoresis and subsequently transferred onto a polyvinylidene fluoride membrane in an ice bath at 100 V for 75 min. Membranes were subsequently blocked with 5% bovine serum albumin (A8020; Solarbio, Beijing, China) at 25 °C for 2 h. Subsequently, the membranes were incubated with a mouse polyclonal antibody against the gemykibivirus Cap protein (prepared in-house) or a mouse anti-β-actin monoclonal antibody (Sino Biological, Beijing, China) overnight. After three washes with tris-buffered saline containing 0.07% Tween-20 (TBST; 10 min/wash), the membranes were incubated with horseradish peroxidase-labeled IgG secondary antibodies (Proteintech, Wuhan, China) at 25 °C for 2 h. After another set of TBST washes (10 min/wash), antibody-reactive bands were visualized using an enhanced chemiluminescence detection system and exposed on a radiographic film.

### 2.6. Immunofluorescence

Cells were seeded on the coverslips previously and fixed with 4% paraformaldehyde (P1110, Solarbio, Beijing, China) for 15 min, permeabilized in 0.1% Triton X-100 (T8200, Solarbio, Beijing, China), and incubated with the primary antibody (targeting the gemykibivirus, prepared in-house) at 4 °C overnight. After three washes with TBST, the cells were labeled with the goat anti-Mouse IgG (H+L) cross-adsorbed secondary antibody (Invitrogen, A-11001) for 1 h at 37 °C. The cells were washed subsequently and stained with 4′,6-diamidino-2-phenylindole (P36971, Invitrogen, Carlsbad, CA, USA). Finally, the cells were analyzed via confocal microscopy (Olympus, Tokyo, Japan).

### 2.7. Transcriptional Analysis by Quantitative Reverse Transcription Polymerase Chain Reaction (qRT-PCR)

Total RNA isolated from the HEK293T cells was treated with TRIzol as described above. cDNA synthesis was conducted using the HiScript II Q RT SuperMix for qPCR (Vazyme Biotech Co., Nanjing, China). Sequence-specific primers for randomly selected significant differential genes were retrieved from the GETprime database https://getprime.epfl.ch/ (accessed on 20 June 2024). The ChamQ Blue Universal SYBR qPCR Master Mix (Vazyme Biotech Co., Nanjing, China) and AceQ Universal U+ Probe Master Mix V2 (Vazyme Biotech Co., Nanjing, China) was used to perform qRT-PCR according to the manufacturer’s protocol. See Table 1 for reference primer sequences.

### 2.8. mtDNA Isolation and Analysis

The HEK293T cells were resuspended in 400 μL buffer containing 150 mM NaCl, 50 mM HEPES pH 7.4, and 20 μg/mL digitonin (Beyotime) to quantify the cytosolic mtDNA release following gemykibivirus infection. The homogenates were incubated on an end-over-end rotator for 10 min on ice, and subsequently centrifuged at 940× *g* for 3min. The centrifugation step was repeated three times. The supernatants were transferred to new centrifuge tubes after each centrifugation step. The final supernatant was centrifuged at 17,000× *g* for 25 min at 4 °C. The supernatant was diluted 10 times and subsequently used for qRT-PCR. Total DNA was isolated from whole-cell extracts using the TIANamp Genomic DNA Kit (TIANGEN, Beijing, China). Quantitative PCR was conducted on pure cytosolic fractions and whole-cell extracts using mtDNA primers. Relative cytosolic mtDNA levels were normalized to the total amount of GAPDH DNA. The qRT-PCR reagents and conditions were the same as those previously. The control group samples were treated using the same procedure as the experimental group.

### 2.9. Intracellular ROS Production Assays

The HEK293T cells were treated as indicated and incubated with serum-free media containing 10 μM 2′,7′-Dichlorodihydrofluorescein diacetate (Beyotime, Shanghai, China) at 37 °C for 20 min. They were subsequently inverted and mixed every 3–5 min to ensure thorough contact between the probe and cells. The cells were then washed with PBS three times and resuspended in PBS at a 2 × 10^7^ cells/mL concentration. Then, the suspension was added to a black, opaque 96-well plate, with three replicates per sample and 100 μL of cell suspension per well. The fluorescence intensity was measured at 488/525 nm using a Synergy H1 multimode microplate reader (Biotek, Shoreline, WA, USA).

### 2.10. Apoptosis Assay

Apoptosis was evaluated using the Annexin V-FITC Apoptosis Detection Kit (Elabscience, Wuhan, China). HEK293T cells were collected, resuspended in 1 × Annexin V Binding Buffer and stained with Annexin V-FITC and PI for 15 min at 25 °C under dark conditions. The cells were subsequently acquired using a FACSCanto II flow cytometer (BD Biosciences, Franklin Lakes, NJ, USA) and analyzed with FlowJo.

### 2.11. Statistical Analyses

Data were analyzed using GraphPad Prism (version 8.0) and are presented as mean ± standard deviation (SD) from three independent experiments. Differences between two groups were assessed using unpaired Student’s *t*-test. Statistical significance was set at * *p* < 0.05; ** *p* < 0.01; *** *p* < 0.001.

## 3. Result

### 3.1. Gemykibivirus Generation Using Reverse Genetics

The HEK293T cells were transfected with gemykibivirus infectious clones, and samples were collected 48 h post-transfection for subsequent analysis. PCR amplification targeting the Cap gene was performed using the DNA extracted from the supernatants of the transfected cells to confirm the presence of viral genomic DNA. The expected amplicons were detected using 1% agarose gel electrophoresis (Figure 1A). Western blot analysis was conducted to examine the expression of viral Cap proteins. A specific band corresponding to the expected molecular weight of the Cap protein in the lysates of transfected cells was found, whereas no signal was observed in the mock-transfected controls (Figure 1B). Additionally, immunofluorescence staining showed a distinct cytoplasmic localization of the Cap protein (Figure 1C). Subsequently, we evaluated the copy numbers of DNA and RNA produced by gemykibivirus after transfection. Viral particles were detectable in the culture supernatant as early as 12 h post-infection (Figure 1D,E). The dynamic change in viral DNA distribution between cells and supernatants suggests a possible balance between genome replication and virion release, indicating that gemykibivirus can be successfully rescued through reverse genetics.

### 3.2. Analysis of the DEGs

The expression values of genes were calculated based on read counts using featureCounts to explore the changes in gene expression profiles induced by gemykibivirus infection. The foldchange values of the genes were calculated using DESeq2. The threshold values *p*-adjust ≤ 0.05 and |Log_2_FC| ≥ 1 were used to identify the DEGs. In total, 1732 unique DEGs were identified following gemykibivirus infection compared to controls, including 640 upregulated and 1092 downregulated genes. A volcano plot was generated to visualize the DEGs distribution (Figure 2), underscoring significant alterations in expression.

### 3.3. Functional Annotation Analysis of the DEGs

Functional annotation and classification were performed by comparing the sequences with the GO and KEGG databases to describe the functions and pathways of the genes obtained from RNA-seq. GO terms and KEGG pathways that satisfy the corrected *p*-value ≤ 0.05 were considered significantly enriched. Several biological processes (BP), including response to hypoxia and chaperone-mediated protein folding were significantly enriched. In the cellular component (CC) category, enriched terms such as transporter complex and transmembrane transporter complex indicated potential changes in the intracellular transport systems. Molecular function (MF) analysis demonstrated significant enrichment in the DNA-binding transcription activator activity and monatomic ion channel activity, underscoring potential alterations in gene regulation and ion transport dynamics (Figure 3A). Some pathways, such as MAPK, TGF-β, and Hippo signaling pathways, were identified as the response mechanisms of canonical pathways activated in the cells during viral infection. Several other pathways have been associated with virus-induced diseases, including prion disease, Parkinson’s disease, transcriptional misregulation in cancer, and oxidative phosphorylation (Figure 3B).

### 3.4. Validation of RNA-Seq Analysis by qRT-PCR

We randomly selected several genes, namely HSPA1A, HSPA1B, JUN, PLA2G4B, PLA2G4A, ELK4, MEF2C, and CACNB4, for qRT-PCR analysis to validate the reproducibility and repeatability of the DEGs identified from transcriptome sequencing (Figure 4). These genes were significantly differentially expressed based on RNA-seq, suggesting the reliability of the DEGs obtained from transcriptome sequencing.

### 3.5. Gemykibivirus Induced mtDNA Release and Enhanced the ROS Levels

Mitochondria are essential for central metabolic pathways and are embedded in intracellular signaling networks that regulate different cellular functions. RNA-seq analysis and qRT-PCR showed a substantial increase in the relative mtRNA expression in the HEK293T cells after gemykibivirus infection (Figure 5A). We quantified the mtDNA levels to further evaluate mitochondrial involvement and observed a considerable elevation compared to the control group (Figure 5B). Additionally, we detected mtDNA release into the cytoplasm, indicating its release from the mitochondrial (Figure 5C). We measured ROS levels in the HEK293T cells to investigate whether the release of mitochondrial DNA contributes to mitochondrial dysfunction. Confocal microscopy revealed increased intracellular ROS in gemykibivirus-infected cells compared to the control group (Figure 5D). Quantification of ROS using a microplate reader further confirmed these results, showing significantly elevated ROS levels in the gemykibivirus group (Figure 5E).

### 3.6. Gemykibivirus Induced Late Apoptosis

We determine the role of the gemykibivirus infection in triggering apoptosis in the HEK293T cells. The isolated cells were labeled with Annexin V-FITC and PI and subjected to an apoptosis assay using flow cytometry. The percentage of late apoptotic cells reached 25.7% (Figure 6A), whereas in the control group, only 15.3% of the cells were in late apoptosis. Apoptosis was evaluated in three independent samples (bar graph, Figure 6B). To further substantiate the induction of cell death by gemykibivirus infection, cell viability was evaluated using the CCK-8 assay (Figure 6C). Consistent with the flow cytometry findings, infected HEK293T cells exhibited significantly reduced viability.

## 4. Discussion

Gemykibivirus has been detected in different human samples, including feces, plasma, serum, blood, pericardial fluid, cerebrospinal fluid, and sputum. Additionally, most Genomoviridae family viruses identified in human samples belong to this virus. Given the high recombination rates and rapid evolution of single-stranded DNA viruses, gemykibivirus pose a potentially considerable threat to humans [16,17]. However, the pathogenic mechanisms underlying gemykibivirus infection remain elusive. RNA-seq facilitates the quantitative analysis and comprehensive evaluation of the transcriptome at the cellular level, aiding in a more precise investigation of gene expression dynamics after gemykibivirus infection. This approach has accelerated research on transcriptional profiles in infected cells, providing valuable insights into virus–host interactions and cellular responses.

To assess how gemykibivirus interacts with HEK293T cells, we analyzed its expression and propagation after transfection. We observed that following transfection, viral DNA and RNA were detectable in both cells and culture supernatant, and the viral load in the supernatant steadily increased. These results indicate that gemykibivirus not only transcribes within the cells, but also produces progeny virus through its own machinery. It is noteworthy that gemykibivirus exhibited relatively high copy numbers as early as 12 h post-transfection, likely due to short-term expression initiated by the transfected circularized genome. However, viral load at 24 and 48 h did not increase exponentially, suggesting that replication may be limited in HEK293T cells, which may not represent the virus’s optimal host environment. In nature, humans are thought to be the natural hosts of gemykibivirus, although the virus has so far been detected only in select body fluids [9,18]. Therefore, we speculate that there may be cell types more permissive to gemykibivirus replication. Accordingly, we simulated a more effective infection by transfecting cells with the gemykibivirus genome to examine host antiviral responses.

In this study, DEG profiles from the HEK293T cells were investigated using various bioinformatics approaches, revealing the dynamic and robust nature of the inflammatory response and mitochondrial dysfunction caused by gemykibivirus infection. This infection upregulated certain gene expressions, such as ATP5PF, ATP6, COX1, MT-ND5, CYTB, and HIF-1α and downregulated CACNA1B and RYR3 expression. CACNA1B and RYR3 are the key calcium ion channel genes that play essential roles in cellular signaling and calcium homeostasis [19,20]. CACNA1B encodes an N-type voltage-gated calcium channel, mainly mediating calcium influx in the neurons and other cell types [21]. Contrastingly, RYR3 functions as a calcium release channel situated in the endoplasmic and sarcoplasmic reticulum, regulating intracellular calcium stores and influencing muscle contraction, neuronal activity, and various cellular processes [22]. Calcium ions act as crucial signaling molecules during mitochondrial oxidative phosphorylation (OXPHOS) [23]. CACNA1B and RYR3 downregulation may disrupt intracellular calcium signaling, which may change the respiratory rate, adjust OXPHOS to meet energy demands, and potentially elevate ROS generation [24]. Meanwhile, the upregulation of all 13 mitochondrial DNA-encoded mRNAs, along with multiple nuclear-encoded genes link to the respiratory chain, such as ATP12A, ATP5MF, ATP5PF, and HIF-1α, suggests a virus-induced mitochondrial functional remodeling following gemykibivirus infection. These results indicate that viral infection may reshape cellular metabolism and influence antiviral responses [25,26].

Mitochondria serve as the central hub of cellular energy metabolism, and their functional changes may be closely associated with viral replication. Several viruses hijack host mitochondrial ATP production systems to fuel replication and protein synthesis [27]. Additionally, increased activity of the mitochondrial electron transport chain may elevate ROS levels and influence host immune responses [28]. Previous studies have suggested that ROS can play dual roles during viral infection, either enhancing host antiviral defenses or being exploited by the virus to facilitate replication and dissemination [29]. Additionally, mitochondria have a substantial effect on apoptosis and autophagy [30]. Some viruses modulate mitochondrial signaling pathways to suppress host cell apoptosis, thereby prolonging cell survival and supporting viral propagation [31]. Mitochondrial gene upregulation and high ROS levels as the gemykibivirus infection progresses may reflect viral regulation of host cell viability.

Hypoxia Inducible Factor 1 Subunit Alpha (HIF-1α), the key functional subunit of HIF-1, acts as one of the crucial cellular targets of endogenous ROS, which modulates stress response gene expression link to inflammation, metabolism, oxygen delivery, and cell survival [32]. HIF-1α upregulation observed might be attributed to ROS produced by the cell during the gemykibivirus infection [33]. This mechanism has also been reported in other viruses, such as influenza virus (H1N1) infection and SARS-CoV. H1N1 infection triggers glycolysis in A549 cells with notable HIF-α upregulation and leads to mitochondrial impairment and heightened ROS generation [34]. HIF-1α’s high expression promotes and aggravates inflammatory responses during the SARS-CoV infection. Furthermore, it can set off the coagulation cascade that can result in in situ pulmonary thrombosis and microclots [35,36]. Gemykibivirus infection results in the enrichment of the pathways linked to hypoxia response and oxidative stress. Given that HIF-1α is a critical regulator of glycolysis, mitochondrial function, and inflammatory responses, its upregulation may contribute to viral pathogenesis by promoting and modifying the metabolic state that supports viral replication while simultaneously influencing host immune signaling.

One limitation of this study was that we did not establish a direct relationship between gemykibivirus infection and mitochondrial dysfunction, ROS production, and apoptosis, nor did we determine whether mitochondria generated ROS. Future studies should explore the specific mechanisms underlying mitochondrial functional alterations in response to gemykibivirus infection. Investigations into ATP production, ROS levels, mitochondrial membrane potential, and apoptosis or autophagy related signaling pathways could provide deeper insights into virus–host interactions [37,38,39]. These results may contribute to the development of novel antiviral strategies targeting mitochondrial function.

Overall, our study demonstrates that transcriptional response of HEK293T cells during gemykibivirus infection. The results further show that gemykibivirus infection could induce cell apoptosis and upregulate the expression of mitochondrial RNA and DNA, leading to mitochondrial dysfunction. These findings provide insights into the differential gene expression between control and the gemykibivirus infected cells; however, the mechanisms underlying needs further study.

## 5. Conclusions

This study demonstrates that gemykibivirus is capable of replicating in HEK293T cells and reveals the transcriptional and antiviral responses during infection, highlighting a potential link between gemykibivirus infection and mitochondrial dysfunction. These findings suggest that this emerging virus may disrupt host cellular homeostasis by targeting mitochondrial pathways. As a newly identified member of the Genomoviridae family that is increasingly detected in human clinical samples, Gemykibivirus deserves greater attention in the study of viral pathogenesis. We identified candidate genes linked to processes such as the immune response, mitochondrial dysfunction, and oxidative phosphorylation, revealing molecular mechanisms of infection. Overall, our findings offer novel insights into the complex interactions between viruses and host cells, and lay a foundation for future investigations into the mitochondrial pathways involved in antiviral responses and viral pathogenesis.

## Figures and Tables

**Figure 1 viruses-17-01331-f001:**
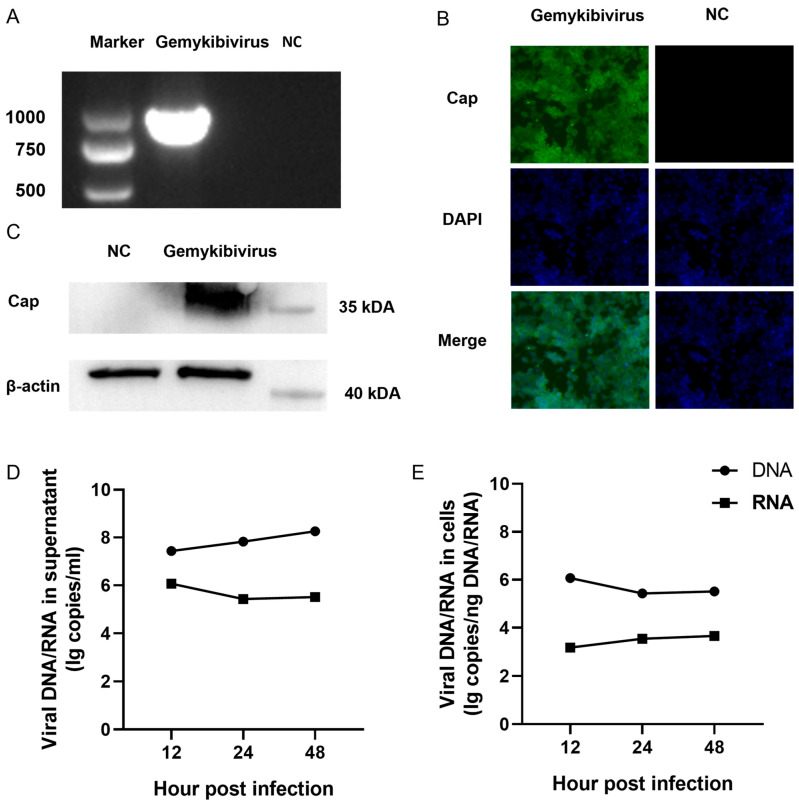
Gemykibivirus could be rescued via transfection of the gemykibivirus infectious clone into the HEK293T cells. (**A**) The cap gene is detected via polymerase chain reaction in the supernatant of the HEK293T cells with 1% agarose gel electrophoresis. Immunofluorescence (**B**) and Western blot (**C**) analysis of the Cap protein expression in the HEK293T cells during infection. Viral DNA and RNA copy numbers were quantified by RT-qPCR at 12, 24, and 48 h post-infection (hpi) in the culture supernatant (**D**) and in the cells (**E**). Note: The complete gel and blot images are provided in the Appendix A, respectively.

**Figure 2 viruses-17-01331-f002:**
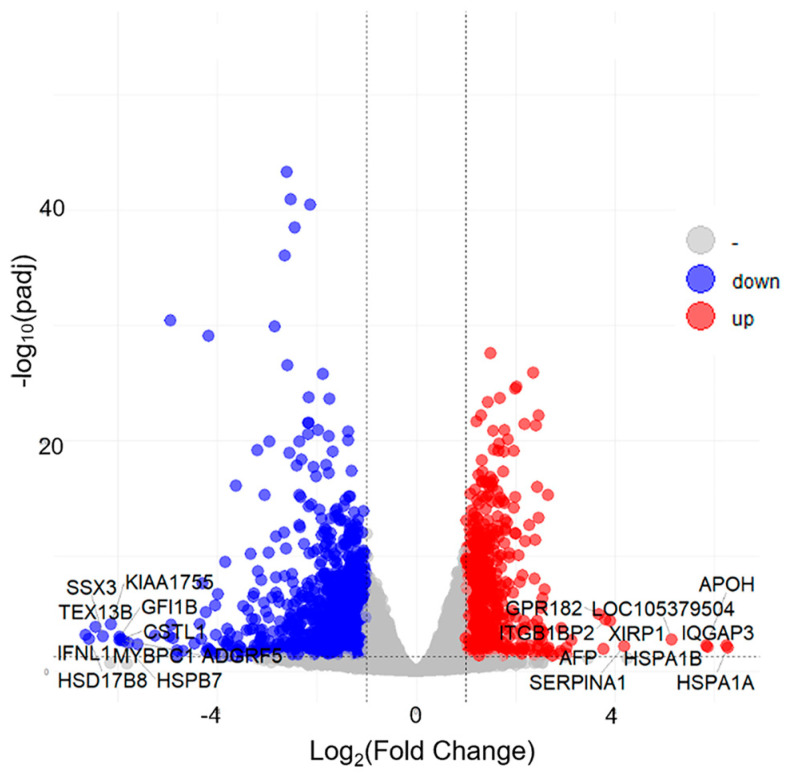
Volcano plot of the differential gene profiling in the HEK293T cells infected with gemykibivirus. Red dots (up), Blue dots (down) and gray dots represent significantly upregulated genes, significantly down-regulated genes, and insignificantly differentially expressed genes, respectively.

**Figure 3 viruses-17-01331-f003:**
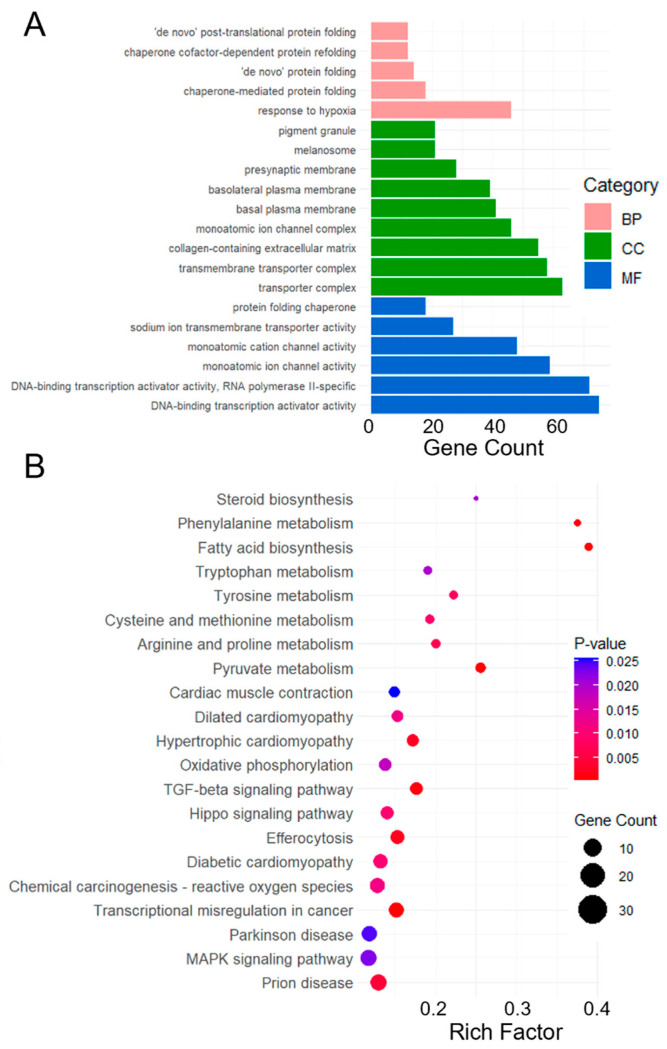
Functional analysis of DEGs of the HEK293T cells in gemykibivirus infection. Here, (**A**) Each BP, CC and MF category is represented by a pink, green, and blue bar, respectively. The height of the bar represents the number of IDs in the user list and also in the category. KEGG pathways enrichment analysis of DEGs (**B**), the *x*-axis represents the rich factor (Ratio of DEGs to total genes in the pathway), *y*-axis lists the significantly enriched pathways. The color of the dots indicates the *p*-value, while the size represents the number of DEGs involved in each pathway.

**Figure 4 viruses-17-01331-f004:**
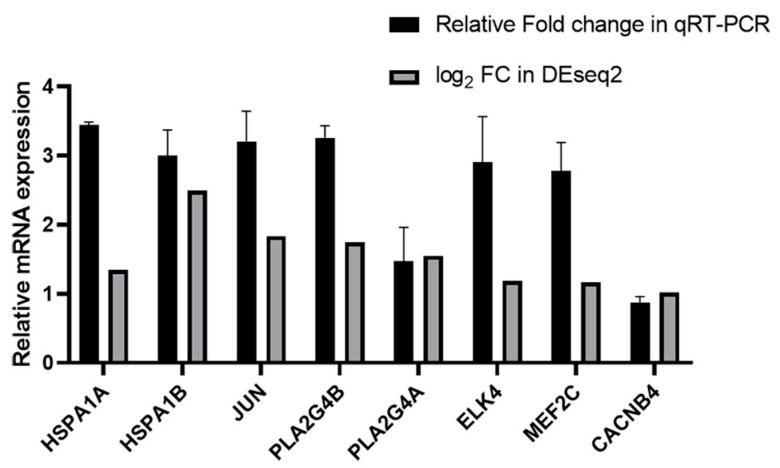
Relative mRNA expression of HSPA1A, HSPA1B, JUN, PLA2G4B, PLA2G4A, ELK4, MEF2C and CACNB4 in the gemykibivirus infection HEK293T cells in qRT-PCR and RNA-seq analysis.

**Figure 5 viruses-17-01331-f005:**
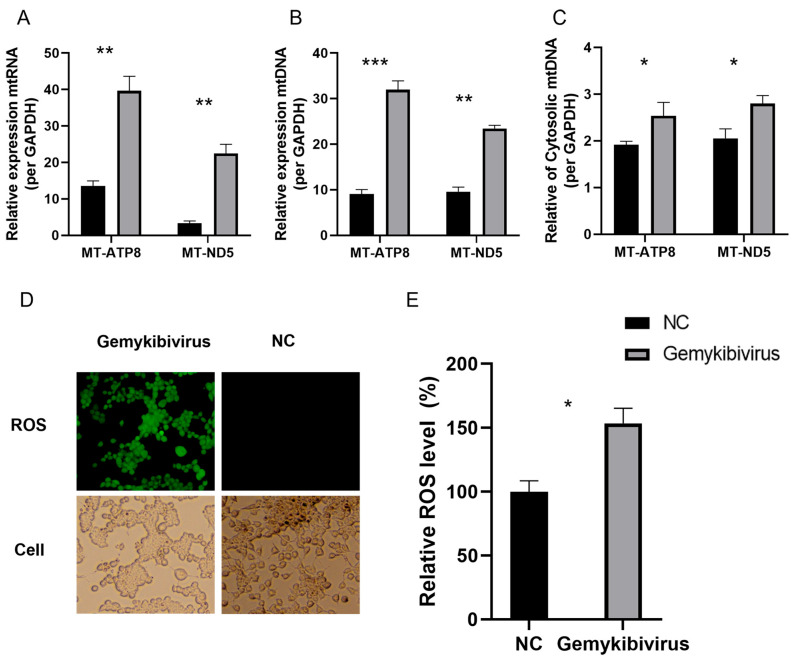
Relative expression levels of the mitochondrial RNA (**A**) and mitochondrial DNA (**B**) in the HEK293T cells following gemykibivirus infection, as well as mitochondrial DNA release into the cytoplasm (**C**). Fluorescence microscopy (**D**) and microplate reader quantification (**E**) of intracellular reactive oxygen species (ROS) in cells stained with a ROS detection kit. The results shown are the means with SDs from three independent experiments. The asterisks indicate statistically significant differences between groups, as assessed by Student’s *t*-test (* *p* < 0.05; ** *p* < 0.01; *** *p* < 0.001).

**Figure 6 viruses-17-01331-f006:**
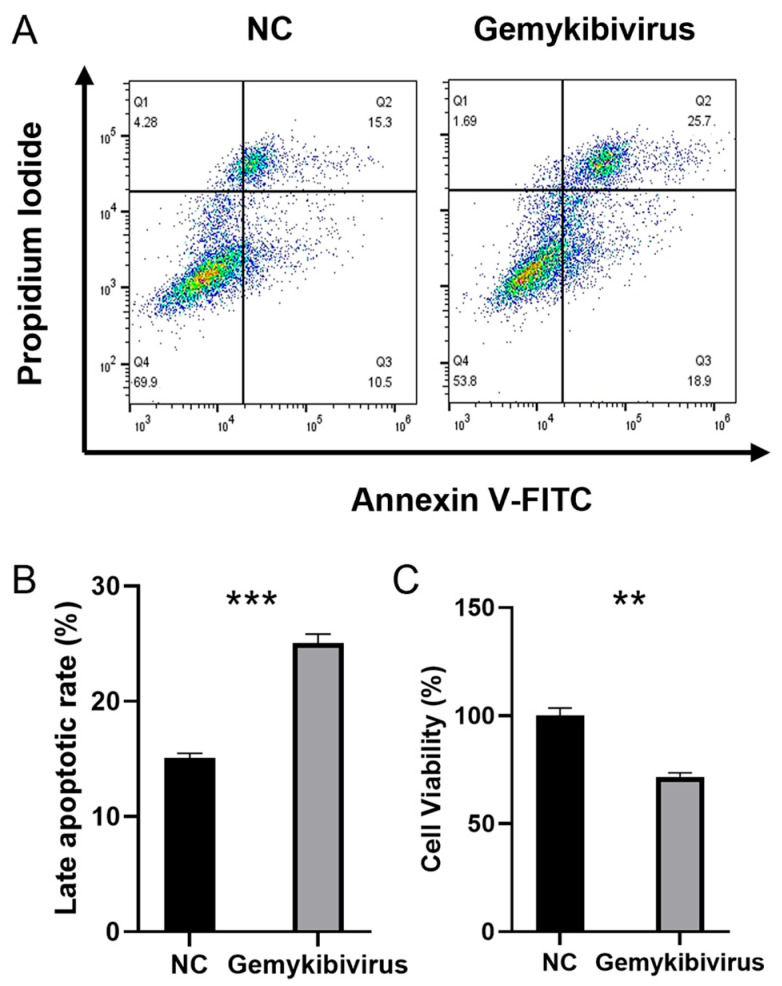
Flow cytometry of the Annexin-V/PI-stained cells was used to determine apoptosis induced by gemykibivirus infection; Q1: necrosis, Q2: late apoptosis, Q3: early apoptosis, and Q4: viable cells (**A**). Quantification of the late apoptotic HEK293T cells (**B**). Cell viability was further evaluated by CCK-8 (**C**). The asterisks indicate statistically significant differences between groups, as assessed by Student’s *t*-test (** *p* < 0.01; *** *p* < 0.001).

**Table 1 viruses-17-01331-t001:** List of primers used in the analysis for qRT-PCR.

Gene	5′–3′	Sequence	Size (bp)
HSPA1A	Forward	CAAGGCCAACAAGATCACC	80
	Reverse	CTCGATCTCCTCCTTGCTC	
HSPA1B	Forward	GAGACCAAGGCATTCTACC	60
	Reverse	TCTCCTTCATCTTGGTCAGC	
JUN	Forward	CAACATGCTCAGGGAACAG	60
	Reverse	ACTGTTAACGTGGTTCATGAC	
PLA2G4B	Forward	ATAATTTCCTGCGTGGCCT	80
	Reverse	CCAGAGTGGTAGCTTTCCA	
PLA2G4A	Forward	AGTATTCCCACAAGTTTACGG	80
	Reverse	GAGTATCAAGCATGTCACCA	
ELK4	Forward	ACTCTCAGTCCTGTTGCTC	80
	Reverse	GTTCAGTACAGAAGGAAACTGG	
MEF2C	Forward	GAACGTAACAGACAGGTGAC	92
	Reverse	CGCAATCTCACAGTCACAC	
CACNB4	Forward	CTCTTTGGAAGAGGACCGG	85
	Reverse	CAGGTTTGGACTTTGCTCTC	
GAPDH	Forward	TCAAGATCATCAGCAATGCC	80
	Reverse	CGATACCAAAGTTGTCATGGA	
Gemykibivirus	Forward	AGNATGTGTATCGCGTCATTT	93
	Reverse	GTACCCGGACNAACCTCTTATC	
	Probe	ATGCCNGTCGAGATNAAGCGTTCC	

## Data Availability

The RNA-seq data have been deposited in the NCBI Sequence Read Archive (SRA) under accession number PRJNA1265676. The reviewer access link is: https://dataview.ncbi.nlm.nih.gov/object/PRJNA1265676?reviewer=nbfof4k77r5h8qrglsm61eq2nt (accessed on 29 September 2025).

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
