# Peer review of "Transcriptome Analysis Reveals Gemykibivirus Infection Induces Mitochondrial DNA Release in HEK293T Cells"

_viruses, 2025, doi:10.3390/v17101331_

Round 1
Reviewer 1 Report (Previous Reviewer 1)
Comments and Suggestions for Authors
The authors considerably improved their manuscript to make it intelligible.
Author Response
Comments: The authors considerably improved their manuscript to make it intelligible.
Responses: We sincerely thank the reviewer for acknowledging our efforts to improve the manuscript. We greatly appreciate your constructive comments, which helped us to enhance the clarity and quality of our work.
Reviewer 2 Report (Previous Reviewer 2)
Comments and Suggestions for Authors
In this manuscript, the authors and their colleagues transfected an infectious clone of gemykibivirus into HEK293T cells, confirming that the virus can be expressed and complete replication in HEK293T cells. They then performed RNA-seq to analyze the changes in the host transcriptome during infection. The authors further found that gemykibivirus infection may lead to mitochondrial dysfunction and induce cell apoptosis, phenomena that are commonly observed during infections by other viruses. This provides a new insight into gemykibivirus, a newly emerging virus. There are several suggestions as follows before publication:
- The authors listed the primer sequences for gemykibivirus. We noticed that a probe was also used, but it was not described in the corresponding Methods section.
- The authors presented the expression and replication of gemykibivirus following infection in the Results section. Since this aspect is particularly important for viral infection process, it is recommended that the authors include a description of viral expression in both the Abstract and Discussion sections.
- Line 260: change HEK293t to HEK293T.
- In “3.1 Gemykibivirus Generation Using Reverse Genetics”, line 185, 195, the authors sometimes use the term ‘transfection’ and other times ‘infection’; the authors are requested to clarify the reason for this inconsistency.
- In “3.5 Gemykibivirus Induced mtDNA Release And Enhanced the ROS Levels”, line 259: the authors measured cellular mtDNA levels above, and subsequently assessed ROS levels; the connection between these two measurements is somewhat abrupt and could benefit from additional explanation.
- Figure 5: are the expression levels of the two genes measured in the cells consistent with the RNA-seq data?
Author Response
Comments 1: The authors listed the primer sequences for gemykibivirus. We noticed that a probe was also used, but it was not described in the corresponding Methods section.
Response 1: We appreciate the reviewer’s suggestion and have added detailed information in the revised manuscript regarding the methods and reagents used for detecting gemykibivirus nucleic acids.
Comments 2: The authors presented the expression and replication of gemykibivirus following infection in the Results section. Since this aspect is particularly important for viral infection process, it is recommended that the authors include a description of viral expression in both the Abstract and Discussion sections (2. Materials and Methods, Section 2.7, Transcriptional Analysis by Quantitative Reverse Transcription Polymerase Chain Reaction (qRT-PCR), P.4 line 145-146).
Response 2: We thank the reviewer for the valuable suggestion. We have added descriptions in both the Abstract and Discussion sections regarding the expression and replication of gemykibivirus in HEK293T cells.
Comments 3: Line 260: change HEK293t to HEK293T.
Response 3: We appreciate this reviewer bringing this mistake to our attention. We have corrected it.
Comments 4: In “3.1 Gemykibivirus Generation Using Reverse Genetics”, line 185, 195, the authors sometimes use the term ‘transfection’ and other times ‘infection’; the authors are requested to clarify the reason for this inconsistency.
Response 4: We thank the reviewer for this insightful comment. In this manuscript, we used a gemykibivirus infectious clone to simulate viral infection. Therefore, when describing the methodological aspect, we used the term “transfection” to emphasize the entry of the viral genome into the cells. In contrast, when referring to the virus invading cells and undergoing replication, we used “infection” to highlight the subsequent impact of gemykibivirus on the host cells.
Comments 5: In “3.5 Gemykibivirus Induced mtDNA Release And Enhanced the ROS Levels”, line 259: the authors measured cellular mtDNA levels above, and subsequently assessed ROS levels; the connection between these two measurements is somewhat abrupt and could benefit from additional explanation.
Response 5: We thank the reviewer for the valuable suggestion. For the sake of conciseness in the Results section, we only presented the experimental outcomes. The relationship between gemykibivirus-induced mtDNA release and ROS levels has been described in the Discussion section (4. Discussion, P. 11-12, Lines 342-365).
Comments 6: Figure 5: are the expression levels of the two genes measured in the cells consistent with the RNA-seq data?
Response 6: We thank the reviewer for the suggestion. The two genes we examined exhibited the same upregulated trend as observed in the RNA-seq data following gemykibivirus infection. To better highlight the consistent upregulation of both RNA and DNA copy numbers of MT-ND5 and MT-ATP8, this additional result further validates the reliability of our RNA-seq data.
This manuscript is a resubmission of an earlier submission. The following is a list of the peer review reports and author responses from that submission.
Round 1
Reviewer 1 Report
Comments and Suggestions for Authors
In this manuscript, Yang and coworkers exposed the outcome of a transcriptome analysis conducted on HEL293T cells infected with a human Gemykibivirus virus (HuGkv) clone. The authors observed that the virus is capable of altering several pathways, including MAP kinase pathways, chemical carcinogenesis, and oxidative phosphorylation. Upon infection, HEK293T cells were characterized with an upregulation of mitochondrial genes, suggesting that HuGkv is capable of modulating mitochondrial activity, including by promoting mtDNA shedding in the cytoplasm. thereby triggering innate immune antiviral response.
The paper deals with a new domain of human virology. Gemybikiviruses are poorly documented in humans. However, the authors skipped important steps of viral characterization that are problematic to fully understand what they are doing exactly.
The principal problem concerns the system of viral replication used.
The authors only mentioned « An infectious clone of gemykibivirus previously established via molecular biology methods wan used to rescue viruses in HEK293T cells by enabling in vitro gemykibivirus circularization, of Gemykibivirus, restoring its natural conformation. »
This is coming without any reference or further description that are nevertheless mandatory to understand the paper. Indeed, there is no available replication system for HuGkv so far. So the important result of the paper should be the implementation of a replication system and not the transcriptome analysis. What is transfected in HEK293T cells?
What is the « molecular biology methods wan » coming together with an in vitro gemykibivirus circularization (gemybikivirus genome circularization ?) and natural conformation restoration (circularity plus DNA single-strandedness) ?
Before getting interested in the transcriptome, we are interested in the viral cycle. In this regard, the next explanation provided by the authors, « Subsequently, the membranes were incubated with mouse polyclonal antibodies (targeting the gemykibivirus, prepared in-house » is also a matter of amazement. We see maybe one band on the blot, but HuGkv encodes for at least two proteins (Rep and Cap). What is detecting the antibody?
The immunofluorescence is not convincing at all.
What is the amount of viral particles produced in the cell supernatant?
Did the authors try measuring viral RNA expression?
Author Response
Comments 1: The paper deals with a new domain of human virology. Gemybikiviruses are poorly documented in humans. However, the authors skipped important steps of viral characterization that are problematic to fully understand what they are doing exactly.
Response 1: We thank the reviewer for highlighting this important point. We fully agree that Gemykibivirus is a poorly documented virus in humans, and a more comprehensive viral characterization would further strengthen the study. When we obtained the circularized full-length genome of Gemykibivirus, we were concerned that natural entry into cells might not allow efficient replication. Therefore, we chose to rescue the virus by directly transfecting circularized genomes into HEK293T cells to ensure successful rescue and to obtain sufficient virus. Our initial focus was to investigate the impact of gemykibivirus infection on host cells, particularly in relation to antiviral immune responses; therefor, wo employed RNA-seq to analyze change in the host transcriptome during infection. In the revised version (3. Result, Section 3.1, Gemykibivirus Generation Using Reverse Genetics, P. 5-6, Lines 183-206) we have added data showing the intercellular viral load and the release of viral copies into the culture supernatant following gemykibivirus transfection, as well as the production of viral RNA. We trust that the revisions have clarified this issue as requested. Additionally, we have included in the Discussion a description of gemykibivirus expression and self-propagation in HEK293T cells. (4. Discussion, P. 11, Lines 296-309,)
Comments 2: The principal problem concerns the system of viral replication used.
Response 2: We appreciate the reviewer’s insightful comment regarding the viral replication system. We fully acknowledge that there is currently no standardized model for Gemykibivirus replication in mammalian cells, and this represents a limitation of our study. Because natural entry routes and infection efficiency remain poorly defined, we employed transfection of circularized full-length genomes as a practical strategy to ensure successful rescue of the virus. Importantly, we observed not only intracellular viral RNA but also viral genomes released into the culture supernatant, indicating that the signals were derived from replication and virus production rather than from transcription of the input genome alone. While we agree that this system may not fully mimic natural infection, it provides a feasible and reproducible platform to investigate the cellular effects of gemykibivirus. Future work will focus on establishing more physiologically relevant infection models to further validate these findings.
Comments 3: The authors only mentioned « An infectious clone of gemykibivirus previously established via molecular biology methods wan used to rescue viruses in HEK293T cells by enabling in vitro gemykibivirus circularization, of Gemykibivirus, restoring its natural conformation. This is coming without any reference or further description that are nevertheless mandatory to understand the paper. Indeed, there is no available replication system for HuGkv so far. So the important result of the paper should be the implementation of a replication system and not the transcriptome analysis. What is transfected in HEK293T cells?
Response 4: We thank the reviewer for this important comment. We apologize for the lack of clarity in our initial description. In the revised manuscript (2. Materials and Methods, Section 2.1, Cells and Virus, P. 2, Lines 75–83), we have added a more detailed description of the construction of the gemykibivirus infectious clone. The construction method has been revised as follows: The full-length gemykibivirus genome was inserted into the pBluescript vector with EcoRI restriction sites at both termini. The recombinant plasmid was propagated in E. coli DH5α, and the viral genome was subsequently excised from the plasmid by EcoRI digestion and circularized with T4 DNA ligase, The ligation products were then purified using a DNA clean-up kit, yielding a circular double-stranded DNA form of gemykibivirus that closely mimics its natural conformation. Approximately 300 ng of the ligated product (the circularized full-length double-stranded gemykibivirus genome) was transfected into HEK293T cells, while the control group was transfected with an equivalent volume of the buffer used to dissolve the ligated gemykibivirus genome.
Comments 5: What is the « molecular biology methods wan » coming together with an in vitro gemykibivirus circularization (gemybikivirus genome circularization ?) and natural conformation restoration (circularity plus DNA single-strandedness) ?
Response 5: We thank the reviewer for pointing out this inaccuracy in our description. In our study, we employed standard molecular biology methods to obtain the full-length gemykibivirus genome and ligated the terminal sequences to generate a circular DNA form, which closely mimics the natural topology of the viral genome. We apologize for the confusion caused by our previous wording and have revised the text accordingly.
Comments 6: Before getting interested in the transcriptome, we are interested in the viral cycle. In this regard, the next explanation provided by the authors, « Subsequently, the membranes were incubated with mouse polyclonal antibodies (targeting the gemykibivirus, prepared in-house » is also a matter of amazement. We see maybe one band on the blot, but HuGkv encodes for at least two proteins (Rep and Cap). What is detecting the antibody?
Response 6: We thank the reviewer for this insightful comment. We apologize for the lack of clarity in the previous description. The in-house antibody used in our study was specifically raised against the Cap protein of gemykibivirus. Accordingly, the immunoblot is expected to show a single band corresponding to the viral Cap protein. To avoid confusion, we have revised the text (2. Materials and Methods, Section 2.5, Protein Extraction and Western Blot Analysis, P. 3, Lines 123) to clearly specify that the antibody targets the Cap protein. This correction ensures that the experimental design and the interpretation of the blot are more precise and transparent.
Comments 7: The immunofluorescence is not convincing at all.
Response 7: We appreciate your comment. To address this concern, we have optimized the immunofluorescence assay and replaced the images with higher-quality data in the revised version of the manuscript (3. Result, Section 3.1, Gemykibivirus Generation Using Reverse Genetics, P. 5-6, Lines 183-206,).
Comments 8: What is the amount of viral particles produced in the cell supernatant?
Response 8: Thank you for this valuable suggestion. We have now quantified viral DNA expression in HEK293T cells at different time points post-transfection using RT-qPCR. Both cellular and supernatant DNA were analyzed. A standard curve generated from copy number versus Ct values was used to calculate viral DNA copy numbers. The results showed that 12 h post-transfection, viral particles in the culture supernatant reached 10^7.44 copies, and the viral load steadily increased over time, reaching 10^8.3 copies at 48 h. These findings demonstrate that gemykibivirus is not only expressed following transfection but can also replicate in HEK293T cells. The results have been added to the revised Results section and are presented in the updated figure (Figure 1 C, D) (3. Result, Section 3.1, Gemykibivirus Generation Using Reverse Genetics, P. 5-6, Lines 183-206).
Comments 9: Did the authors try measuring viral RNA expression?
Response 9: Thank you for this valuable suggestion. We have now quantified viral RNA expression in HEK293T cells at different time points post-transfection using RT-qPCR. Both cellular and supernatant RNA were analyzed. A standard curve generated from copy number versus Ct values was used to calculate viral RNA copy numbers. The results have been added to the revised Results section and are presented in the updated figure (Figure 1 C, D) (3. Result, Section 3.1, Gemykibivirus Generation Using Reverse Genetics, P. 5-6, Lines 183-206).
Reviewer 2 Report
Comments and Suggestions for Authors
Gemykibivirus, an emerging virus of the recently established genus in the family of Ge- 11
nomoviridae, had been discovered in human blood and cerebrospinal fluid and a variety 12
of other body fluids. However, the molecular mechanisms of Gemykibivirus entrance into 13
the host cells and its pathogenicity remain poorly understood. The authors constructed an infectious clone of Gamakibovirus using molecular biology techniques, rescued the virus in HEK293T cells, and analyzed changes in the host transcriptome during infection via RNA sequencing (RNA-Seq). In the high-throughput transcriptome analysis, a total of 1732 significantly differentially expressed genes were identified. A series of subsequent experiments confirmed that Gamakibovirus can lead to an increase in mitochondrial DNA copy number, promote the release of mitochondrial DNA into the cytoplasm, enhance the production of reactive oxygen species, and trigger other cellular antiviral responses. This lays a foundation for revealing the relationship between Gamakibovirus and human diseases.
This article generally meets the journal's publication requirements, but there are several suggestions before publication:
1、In "3.1. Gemykibivirus Generation Using Reverse Genetics", the description of the "Reverse Genetics" technique is overly simplistic. It is recommended that the authors add the necessary experimental methods in "2. Materials and Methods".
2、In "2.2. Isolation of Total RNA and RNA-seq", the authors mention that "the control group was transfected with 10-15 μL/well of ddH₂O". It is advisable to explain why no negative controls such as "NC (negative control)" or empty vectors were used for transfection.
3、In Figures 3 and 4, the authors verified some DEGs; however, these genes do not have a direct relationship with the "mitochondria-related metabolic pathways" and "apoptotic pathways" verified in the subsequent text. The authors are requested to provide an explanation.
- HSPA1A is a gene associated with the heat shock response, and the protein it encodes plays an important role in cellular stress responses and the maintenance of protein homeostasis.
- HSPA1B is a key member of the heat shock protein 70 (HSP70) family, encoding a stress-inducible heat shock protein that plays a central role in cells' response to various stress stimuli and the maintenance of protein homeostasis.
- UN is an important proto-oncogene, and the protein it encodes belongs to the activator protein-1 (AP-1) transcription factor family, exerting a critical role in multiple biological processes such as cell proliferation, differentiation, apoptosis, and stress responses.
- PLA2G4B is a significant member of the phospholipase A₂ (PLA₂) family, and the protein it encodes plays a key role in processes including lipid metabolism, inflammatory responses, and cellular signal transduction.
- LK4 is a member of the ETS transcription factor family, functioning importantly in cellular signal transduction, regulation of gene expression, and various biological processes.
- MEF2C (myocyte enhancer factor 2C) is a crucial member of the MEF2 family of transcription factors, playing a vital role in embryonic development, cell differentiation, maintenance of tissue homeostasis, and the occurrence of diseases.
- CACNB4 is the gene encoding the calcium channel β4 subunit, and the protein encoded by this gene is an important component of voltage-gated calcium channels (VGCCs), playing a key role in physiological processes such as nerve conduction, muscle contraction, and cellular signal transduction.
- When verifying the "mitochondria-mediated metabolic pathways and cell apoptosis", the authors employed overly single verification methods. To enhance the credibility of the results, multiple verification methods should be used for validation, such as gene expression level analysis, protein translation level detection, confocal microscopy, and transmission electron microscopy.
-
Could the authors use experimental methods such as "knockdown, knockout, or overexpression" to perform necessary in vitro validation of the experimental results?
-
Could appropriate animal models be used for necessary in vivo validation?
Author Response
Comments 1: In "3.1. Gemykibivirus Generation Using Reverse Genetics", the description of the "Reverse Genetics" technique is overly simplistic. It is recommended that the authors add the necessary experimental methods in "2. Materials and Methods".
Response 1: We thank the reviewer for this suggestion. Sorry for our negligence about this section. In the revised manuscript (2. Materials and Methods, Section 2.1, Cells and Virus, P. 2, Lines 75–83), we have added a more detailed description of the construction of the gemykibivirus infectious clone. The construction method has been revised as follows: The full-length gemykibivirus genome was inserted into the pBluescript vector with EcoRI restriction sites at both termini. The recombinant plasmid was propagated in E. coli DH5α, and the viral genome was subsequently excised from the plasmid by EcoRI digestion and circularized with T4 DNA ligase. The ligation products were then purified using a DNA clean up kit, yielding a circular double-stranded DNA form of gemykibivirus that closely mimics its natural conformation.
Comments 2: In "2.2. Isolation of Total RNA and RNA-seq", the authors mention that "the control group was transfected with 10-15 μL/well of ddH₂O". It is advisable to explain why no negative controls such as "NC (negative control)" or empty vectors were used for transfection.
Response 2: We thank the reviewer for the constructive comments, which have helped us improve the manuscript (2. Materials and Methods, Section 2.2, Isolation of Total RNA and RNA-seq, P. 2, Lines 85–87). We have clarified inaccuracies in the original description. In the Gemykibivirus group, HEK293T cells were transfected with the circularized full-length viral genome. However, the original version did not explain the rationale for transfecting the control group with ddH₂O. In addition, we revised the description of the DNA transfected into HEK293T cells, specifying that 300 ng of DNA was transfected per well instead of the previously reported transfection volume, to provide a more accurate description. As the circularized viral genome was used and no empty vector was included, the negative control group was transfected with an equivalent volume of the buffer used to dissolve the viral DNA. This design ensures that any observed effects are attributable to the viral genome itself rather than the buffer.
Comments 3: In Figures 3 and 4, the authors verified some DEGs; however, these genes do not have a direct relationship with the "mitochondria-related metabolic pathways" and "apoptotic pathways" verified in the subsequent text. The authors are requested to provide an explanation.
Response 3: We thank the reviewer for this valuable comment. We apologize for the lack of clarity in our description. In Figure 4 (3. Result, Section 3.4, Validation of RNA-Seq Analysis by qRT-PCR, P. 8, Lines 242–251), several genes were randomly selected for RT-qPCR validation to confirm the reliability of the RNA-seq results. Therefore, these genes may not be directly related to the “mitochondria-related metabolic pathways” or “apoptotic pathways” discussed later. In the Discussion section (4. Discussion, P. 11-12, Lines 314–355), we instead focused on representative DEGs with functional relevance, such as RYR3, CACNA1B and HIF-1α, to infer the potential impact of gemykibivirus infection.
Comments 4: When verifying the "mitochondria-mediated metabolic pathways and cell apoptosis", the authors employed overly single verification methods. To enhance the credibility of the results, multiple verification methods should be used for validation, such as gene expression level analysis, protein translation level detection, confocal microscopy, and transmission electron microscopy.
Response 4: We sincerely thank the reviewer for this constructive suggestion. To strengthen the validity of our findings regarding mitochondria-mediated metabolic pathways and apoptosis, we have performed additional validation experiments. Specifically, we supplemented our study with CCK-8 assays, which demonstrated that gemykibivirus infected HEK293T cells exhibited significantly reduced viability relative to controls, consistent with the flow cytometry results (3. Result, Section 3.6, Gemykibivirus Induced Late Apoptosis, P. 10, Lines 270–283). Furthermore, we included confocal microscopy analysis to visualize the intracellular localization of ROS, thereby providing additional evidence supporting mitochondrial involvement (3. Result, Section 3.5, Gemykibivirus Induced mtDNA Release and Enhanced the ROS Levels, P. 9, Lines 252–269).
We sincerely thank the reviewer again for the constructive comments. The additional experiments you suggested, such as protein translation level detection and transmission electron microscopy, would indeed greatly enhance the credibility of our conclusions. However, we currently lack the necessary resources to perform these analyses. We believe that when we are able to implement these experimental approaches in the future, your suggestions will provide invaluable guidance to strengthen our study.
Comments 5: Could the authors use experimental methods such as "knockdown, knockout, or overexpression" to perform necessary in vitro validation of the experimental results?
Response 5: We sincerely thank the reviewer for this constructive suggestion. At present, research on gemykibivirus is still very limited, and no specific host gene has yet been identified to be directly involved in viral entry or replication. In our study, we rescued gemykibivirus and tested its ability to infect multiple cell lines. Interestingly, we found that HAM cells supported gemykibivirus replication more efficiently, which may be related to the higher expression of potential cellular receptors in this cell type. However, since the exact receptor or host factor has not yet been identified, knockdown, knockout, or overexpression experiments cannot be meaningfully performed at this stage. We fully agree that once such a candidate receptor or host factor is discovered, these genetic approaches will be highly valuable to further validate and extend our findings, and we plan to address this in our future work.
Comments 6: Could appropriate animal models be used for necessary in vivo validation?
Response 6: We sincerely thank the reviewer for the constructive suggestion. We have attempted to use BALB/c mouse models to assess whether gemykibivirus can infect mice. Encouragingly, viral copies were detectable in certain tissues, such as the brain and lung. However, the animal model experiments are not yet fully optimized, and we believe the current data do not meet the standards for inclusion in this manuscript. We plan to further refine the mouse model and conduct additional studies to explore gemykibivirus in vivo in future work.